# How Much Malperfusion Is Too Much in Acute Type A Aortic Dissections?

**DOI:** 10.3390/jcm8030304

**Published:** 2019-03-04

**Authors:** Horea Feier, Dragos Cozma, Marius Sintean, Petre Deutsch, Sorin Ursoniu, Marian Gaspar, Cristian Mornos

**Affiliations:** 1Department of Cardiovascular Surgery, Institute for Cardiovascular Diseases, Str. Gh. Adam nr. 13A, 300310 Timisoara, Romania; msintean@yahoo.com (M.S.); petrudeutsch@gmail.com (P.D.); mariangaspar24@yahoo.ro (M.G.); mornoscristi@yahoo.com (C.M.); 2University of Medicine and Pharmacy Timisoara, Pta. Eftimie Murgu nr.2, 300041 Timisoara, Romania; sursoniu@umft.ro

**Keywords:** aorta, dissection, malperfusion, acidosis

## Abstract

(1) Background: Malperfusion is a central limiting factor in the setting of acute Type A aortic dissections (AAAD). We sought to find preoperative metabolic acidosis thresholds that might influence decision-making in this setting. (2) Methods: We retrospectively reviewed consecutive patients operated on with AAAD between January 2002 and December 2017. We analyzed preoperative variables that might influence early and long-term outcomes, with particular emphasis on malperfusion markers. (3) Results: Our sample consisted of 153 patients, most of them male (69.2%), with a mean age of 55.89 ± 12.8 years. Malperfusion was present in 20.9% of cases: peripheric 25, renal 7, cerebral 4, and mesenteric 3. Cardiogenic shock was present in 18.9% of patients. Logistic regression revealed entry site (odds ratio (OR) = 2.83, *p* = 0.03), cardiogenic shock (OR = 3.30, *p* = 0.03), prebypass pH (OR = 0.93, *p* = 0.02) as independent risk factors for early death (<30 days). Receiver operating characteristic (ROC) analysis identified a prebypass pH of 7.25 as a cutpoint for an unfavourable early outcome. Patients whose prebypass pH was ≤7.25 had a 2.98 higher relative risk (65.7% vs. 22%, *p* < 0.001). Prebypass pH 7.25 (hazard ratio (HR) = 4.00, *p* < 0.01) and entry site (HR = 2.10, *p* = 0.04) were independent predictors of early phase survival (<30 days), while long-term survival (>30 days) was determined by age >65 years (HR = 3.12, *p* = 0.02). (4) Conclusions: Patients with a prebypass pH ≤ 7.25 have an unacceptably high early mortality after AAAD repair. Those patients might benefit from a two-stage approach.

## 1. Introduction

Despite contemporary advances in surgical technique and anesthetic management, mortality in acute Type A aortic dissection (AAAD) remains high, ranging from 15–30% [1,2,3], although there has been a constant trend to improve results over the last 20 years [4,5].

In the last decade, a focus on malperfusion has resulted in the establishment and subsequent validation of the Penn classification [6]. The presence of malperfusion can increase mortality to 29% [7], or even 43% if 3 or more systems are affected [8]. Some recent reports have argued for an “ischemia-first” approach, in which severe malperfusion is addressed prior to aortic repair [9,10,11,12]. On the other hand, the Stanford Group have recently published excellent results using the traditional “aorta-first” technique [13]. 

There are no studies determining the most sensitive marker for preoperative malperfusion. Lactic acid levels above 6 mmol/L [14], as well as base deficit greater than −10 mEq/L [15], have very recently been proposed as cutoff points for an unfavorable outcome, however none of these markers have been analyzed in conjunction with one another, as well as with preoperative pH. 

The purpose of this study was to identify preoperative malperfusion levels that would severely impact early and late survival by using an ‘’aorta-first” technique. 

## 2. Materials and Methods

We retrospectively analyzed all admissions for AAAD in our institution between January 2002 and December 2017. AAAD was defined as an aortic dissection diagnosed less than 14 days after the onset of symptoms. During this period, 163 patients were admitted with the above diagnosis. Five patients were deemed too sick to be operated upon and were treated conservatively. We performed 158 consecutive surgeries for repair of AAAD. Three patients had incomplete preoperative data while two were lost to follow-up. Our final sample included 153 patients.

Patients were diagnosed on the basis of a contrast-enhanced chest computerized tomography (CT) scan and/or transthoracic ultrasonography. On admission, we performed transthoracic echocardiography in all patients and took venous blood samples for complete analysis. On stable patients, an arterial sample for complete blood-gas and acid-base analysis was also harvested on arrival after 2008 (*n* = 84), but not prior to that. These data were not used for this study. Unstable patients were rushed directly to the operating room. A set of blood-gas and acid-base analysis was taken in the operating room in all patients (*n* = 153) and this set of data was used for statistical analysis.

### 2.1. Surgical Technique

Stable patients were peripherally cannulated prior to sternotomy using the axillary or femoral arteries. In unstable patients we performed sternotomy and a limited pericardotomy in order to relieve cardiac tamponade, and cannulated afterwards. Venous inflow was obtained via the right atrium in all patients. The left side of the heart was vented using the right superior pulmonary vein. The patient was cooled to a core temperature of 25 degrees Celsius. The aorta was crossclamped just before reaching this temperature, transected and a dose of cardioplegia was delivered via the coronary ostia. The ascending aorta was inspected for the entry-site of the dissection. The distal anastomosis was performed using either a closed (44.4%), or open distal (55.5%) technique, according to the surgeon’s preference. When performing an open distal anastomosis, cerebral protection was achieved using antegrade cerebral perfusion at a rate of 10 mL/kgc/min. After completing the distal anastomosis systemic perfusion was resumed and the patient was rewarmed in an antegrade fashion using either the previously cannulated axillary artery or a side-branch in the aortic prosthesis. The proximal anastomosis was performed next. The root and aortic valve were assessed. Reasons for performing complete root resection were extensive dissection involving more than the non-coronary sinus, a dilated aortic root (>5 cm) or an abnormal aortic valve. In such cases it was replaced using a conduit graft, root reimplantation or remodeling techniques. The aortic root was treated conservatively, in most cases, by reapparoximating the dissecting layers with surgical glue (Bioglue, Kennesaw, GA, USA) and reinforcing them with generous outer Teflon felts.

### 2.2. Definitions

Peripheral malperfusion was defined as pulse loss, pain and signs of ischemia on inspection and palpation in the affected arm or leg, associated with dissection of the iliac, subclavian or brachiocephalic trunk on CT. Cerebral malperfusion was defined as obnubilation, coma or cerebrovascular accident associated with dissection of the brachiocephalic trunk or left common carotid artery. Mesenteric malperfusion as abdominal pain, distension, hemorrhagic or melenic stool on digital rectal exam in a patient with dissection of the superior mesenteric artery. Renal malperfusion was diagnosed as false lumen compression or dissection of at least one renal artery associated with creatinine rise above 50% of the normal upper limit for age and bodyweight [16]. Shock was defined was a systolic blood pressure <90 mmHg or patient on cathecolamine support on arrival.

### 2.3. Statistical Analysis

Continuous variables were expressed as mean ± standard deviation (SD). Categorical variables were presented as percentages. Student’s *t*-test with or without Satterwaithe’s correction was used on normally distributed continuous variables. Mann–Whitney’s ranksum test was used for the other continuous variables. For categorical variables we employed Fisher’s exact test. Univariate analyses were performed in order to determine the variables associated with early mortality (<30 days). We included only acid-base values taken in the operating room, just prior to incision (the last values of pH and base deficit), as they were available in all patients. Variables that achieved a *p* value <0.2 in the univariate analysis were introduced in a logistic regression model with early death as the dependent variable. Variance inflation factors (VIF) were computed to check for collinearity in the included variables. Receiver operating characteristic (ROC) analysis and curves were constructed, and cutoff values were obtained using the Liu method. The patients were divided into two groups according to this cutoff value. Postoperative complications were compared on these groups. Survival analysis was performed on an early (<30 days) and late (>30 days) hazard phase using Cox regression analysis on selected significant variables. Kaplan–Meier curves were computed. In all cases, a *p* value <0.05 was deemed as statistically significant. 

## 3. Results

There were 106 males (69.2%) and 47 females (30.7%), aged 55.89 ± 12.8 years (range 14–82 years). Risk factors included hypertension (82.8%), diabetes (7.8%), bicuspid aortic valve (9.8%) and obesity (29.4%). Eight cases (5.2%) occurred after previous cardiac surgical procedures: seven aortic valve replacements and one triple coronary artery bypass. One patient, aged 14, had untreated aortic coarctation, while five (3.2%) had Marfan syndrome.

More than a third of the patients (35.9%) had a pericardic effusion ≥10 mm on preoperative transthoracic ultrasonography. Severe aortic incompetence was found in 35.2% of patients. The size of the ascending aorta was 5.2 ± 1.1 cm. Mean ejection fraction was 53.4 ± 6.5%. 

### 3.1. Malperfusion

Global or local malperfusion (Penn nonAa) was present in 57 patients. Global malperfusion with hemodynamic shock was present in 29 patients (18.9%). Six patients were intubated on admission due to severe hemodynamic compromise. Local malperfusion was present in 32 patients (20.9%): peripheric 25, renal 7, cerebral 4 and mesenteric 3. Twenty-six patients (17%) had a single ischemic territory, 5 (3.2%) had two and 1 (0.6%) had three (Appendix A). Three patients were diagnosed with AAAD after embolectomy (2) or iliac PTCA (1) for acute limb ischemia.

### 3.2. Early Results

There were 32% early deaths in our cohort. Of note, the three patients that had preoperative limb ischemia relieving procedures survived the subsequent AAAD repair. Univariate analysis performed on 26 preoperative variables found age> 65 years (*p* < 0.01), prebypass pH (*p* < 0.01), prebypass base deficit (*p* < 0.01), cardiogenic shock (*p* < 0.01), Penn non-Aa (*p* < 0.001), serum creatinine >1.7 mg/dL on arrival (*p* = 0.02), a pericardial effusion >10 mm (*p* < 0.01), ejection fraction (*p* = 0.03) and entry site outside the ascending aorta (*p* = 0.04) as significantly linked to early death (Table 1). Logistic regression performed with early death as the dependent variable, found entry site outside the ascending aorta (odds ratio (OR) = 2.83, 95% CI = 1.07–7.42, *p* = 0.03), cardiogenic shock (OR = 3.30, 95% confidence interval (CI) = 1.10–9.82, *p* = 0.03), prebypass pH (OR = 0.93, 95% CI = 0.87–0.98, *p* = 0.02) as independent predictors of early death (Table 2). Penn non-Aa status was not included in the model because of the strong collinearity between it, cardiogenic shock and malperfusion (VIF = 14.57). The receiver operating curve of the logistic model had an area under the curve (AUC) of 0.78 (Figure 1). The receiver operating curve for prebypass pH was constructed. A cutoff point of 7.25 was obtained for prebypass pH (area under the ROC curve at cutoff 0.68, specificity 0.88, sensitivity 0.47). Patients whose prebypass pH was inferior to 7.25 had a 2.98 times higher relative risk of early death (65.7% vs. 22%, *p* < 0.001). Patients with a low prebypass pH (<7.25) were more likely to be in shock (*p* < 0.01, Spearman’s coefficient = 0.37), have pericardial effusion >10 mm (*p* < 0.01, Spearman’s coefficient = 0.30) but all other preoperative variables were similar, including malperfusion (*p* = 0.8, Spearman’s coefficient = 0.02).

Local malperfusion had a variable effect on preoperative pH according to the location of the ischemia: patients with mesenteric ischemia had the lowest preoperative pH (7.25 ± 0.13), followed closely by cerebral ischemia (7.26 ± 0.18). Peripheral ischemia seemed to be better tolerated (7.32 ± 0.11). 

The lowest preoperative pH was present in the group of patients whose unstable hemodynamic status required them to be intubated before the procedure: 7.08 ± 0.14. None of them survived the operation.

### 3.3. Intraoperative Data

Intraoperative variables between these two groups (preoperative pH ≤ 7.25, yes/no) were similar: crossclamp time (120.35 ± 50.13 vs. 119.77 ± 41.84 min, *p* = 0.95), bypass time (201.77 ± 90.63 vs. 220.11 ± 74.73, *p* = 0.27), circulatory arrest time (35.04 ± 19.05 vs. 34.97 ± 15.97, *p* = 0.32), entry site in the ascending aorta (*p* = 0.62). The only difference was the use of circulatory arrest, which was more prominent in patients with a low preooperative pH (74.2%, vs 50%, *p* = 0.01). These two groups exhibited similar postoperative complications, with the exception of acute renal failure (14% vs. 37.9%, *p* < 0.01). Their length of stay in the intensive care unit (ICU) was similar (6 ± 7.56 vs. 5 ± 5.63, *p* = 0.51). Four patients had in addition to AAAD repair, concomitant limb revascularization procedures: three femoro-femoral bypasses and one Fogarty catheter embolectomy.

### 3.4. Long-Term Results

Long-term follow-up was 3.42 ± 3.92 years (maximal survival 14.95 years) (Figure 2). Survival at 1, 3 and 5 years was 61.33 ± 3.98, 58.32 ± 4.06 and 56.03 ± 4.21, respectively. Early survival (<30 days) was determined by prebypass pH ≤ 7.25 (hazard ratio (HR) = 4.00, 95% CI = 1.64–9.75, *p* < 0.01) and entry site outside the ascending aorta (HR = 2.10, 95% CI = 1.00–4.42, *p* = 0.04) (Table 3). Cox survival analysis censored at 10 years of follow-up revealed that long term hazard (>30 days) was determined by age> 65 years (HR= 3.12, 95% CI = 1.17–8.31, *p* = 0.02) (Table 4, Figure 3).

## 4. Discussion

The treatment of AAAD has undergone refinements that have allowed surgical mortality to be lowered constantly to around 22–24% [4,9]. 

There is an increased focus on preoperative malperfusion as a major determinant of mortality in AAAD. In this setting, there are currently two differing attitudes: the traditional “aorta-first” philosophy, in which the AAAD patient is rushed to surgery as quickly as possible after the initial diagnosis in order to prevent catastrophic rupture or death by cardiac tamponade, with the reasoning that any malperfusion will subside once the entry site was been resected. On the other hand, the “ischemia-first” attitude will seek to restore blood flow in the affected territories prior to performing aortic surgery as they argue that severe, unchecked malperfusion will only worsen after performing open distal anastomosis in circulatory arrest. This technique, on the other hand, may dramatically increase the interval between patient admission and aortic repair and thus increase mortality up to 33% [10].

The Cornell [17] and, more recently, Stanford groups [13] are proponents of the first attitude. Girardi et al report spectacular results, with mortality as low as 4.4% in the malperfused group vs. 5.1% in the normal group [17]. Their definition of malperfusion included central nervous system, visceral, renal and peripheral territories. The mortality in the Stanford series is similar in patients without malperfusion (8.5%), but somewhat higher (13.4%) in patients with malperfusion. Their definition of malperfusion excluded the central nervous system and included visceral, peripheral and renal territories. No patients were treated pre-operatively for malperfusion. Their mid-term outcomes revealed more branch revascularization procedures in the malperfusion group, which were performed after a median delay of 2 days.

In contrast with the above, proponents of the “ischemia-first” attitude argue that severe, unbalanced preoperative malperfusion and ischemia will become irreversible should the patient be operated for replacement of his diseased aorta. The trade-off in this situation is a longer time-to-aortic-repair, that might increase mortality. Patel et al. had a strategy of resolving the local malperfusion syndromes by interventional techniques (fenestration and/or stenting) and operate on the aorta after a median of 2 days [10]. Their mortality in patients with malperfusion reached, however, 33% (23 of 70), before attempted aortic repair. At the same time, their cohort included an unusually low number of patients in cardiogenic shock on admission, only 5.2%, and this undoubtedly afforded them the necessary time to treat those patients by interventional means. A more recent study by Uchida et al. underscores that strategy [12]. They presented their series of early reperfusion in a mixed sample of acute (69%) and chronic (31%) aortic dissections, only 81% of which underwent surgical repair. They included patients with coronary, cerebral, visceral and peripheral malperfusion in their definition. Renal malperfusion was not included. Reperfusion was undertaken preoperatively or intraoperatively by PTCA, surgical fenestration or direct perfusion. Mortality in the central repair group was significantly higher if preoperative malperfusion was not addressed by early reperfusion (18%). On the other hand, early reperfusion reduced mortality to very low levels (3.4%), which was similar to the one in the no malperfusion group [12].

At the same time, there is much variation in the definition of organ malperfusion. Some authors include renal and cerebral malperfusion in the model [8,10,18] while others do not [7,15] and this may explain the uneven results in the literature.

The merit of the Penn classification has been to unify the concepts of global (cardiogenic shock) and local ischemia (malperfusion) [6,19]. Most studies agree with the severe impact on early and late mortality in Penn nonA patients [7,8,14,15,18]. 

Which markers of malperfusion should we assess? Lawton et al. included the lowest preoperative pH and base deficit values in the statistical model [15] and this value of pH was not significant. The reason for this may have been the fact that the lowest preoperative pH and base deficit were calculated in some patients just prior to surgical intervention (as in our series) and not modified while in others at an earlier time, and thus were subsequently modified by buffering. Bennett and coworkers evaluated, similar to us, the closest to intervention measured serum lactate value, but not pH or base deficit [14]. They found 6.0 mmol/L and, respectively, 6.9 mmol/L as cutoff values for 30-day and 1-year survival. 

Our sample consisted of almost all patients admitted in our institution with AAAD over a 15-year period. Only five cases were deemed too sick to be treated surgically and this attitude may have contributed to our mortality (32.03%), which is higher than reported in the IRAD database [4]. Mortality was higher at the beginning of our experience, between 2002–2007, 38.71%, as we operated on all AAAD patients referred to us, regardless of their hemodynamic status, even those with cardiac arrest and a preoperative pH as low as 6.89. The rate of patients in cardiogenic shock was high in our sample (18.95%) and 35.95% patients had a preoperative pericardial effusion >10 mm which is why most of the metabolic acidosis was due to global ischemia. The six patients who were intubated on admission due to severe hemodynamic compromise did not survive the procedure. However, all three patients who were diagnosed with AAAD after a previous embolectomy or PTCA survived the subsequent AAAD repair. Stable patients should have their preoperative acid-base status optimized by improving cardiac output, diuresis, and buffering prior to surgery. We believe the prebypass pH to be the net result of these attempts and, as our results show, entering the operation with a pH ≤ 7.25 would severely impact short-term survival, as compensation mechanisms are insufficient. Global ischemia due to cardiogenic shock seems to have a bigger impact than local ischemia and these patients should be operated upon as quickly as possible if their preoperative pH allows. On the other hand, patients with local malperfusion syndromes and a pH > 7.25 might benefit from a two-stage approach: an interventional/surgical reperfusion in the affected territory followed by a subsequent central repair. 

While entry site and severe acidosis would limit early survival, they did not influence the late hazard phase (>30 days) in our sample. It seems that after the acute event, age >65 years is the most important predictor of long-term survival. The preoperative ejection fraction showed a trend to influence long-term outcome (*p* = 0.08), but there were insufficient patients with a preoperative EF <40% in order to lend weight to this assumption. 

### Limitations

Our study was retrospective and as such has all the limitations of this type of research. A prospective one would better define which factors, measured at the time of admission, are modifiable and to what extent. Data on admission pH was available only in about half the patients and although they seem to show a similar predictive power for early death as prebypass pH, a larger sample is needed to verify this, as some of the patients will improve their admission malperfusion status by medical treatment. Another critique could be that intraoperative variables were not taken into account and some of them might independently impact survival (in particular bypass time and open distal anastomosis). To account for such factors would have needed a much larger patient cohort for propensity matching to be performed and still get a sample large enough to perform multivariable analysis confidently.

## 5. Conclusions

Severe preoperative malperfusion has a marked effect on early survival after AAAD, but did not influence long-term outcome. Preperative metabolic acidosis should be compensated for as much as the hemodynamic status allows by medical, interventional or surgical techniques as entering the operation with a pH ≤ 7.25 would result in an unacceptably high mortality.

## Figures and Tables

**Figure 1 jcm-08-00304-f001:**
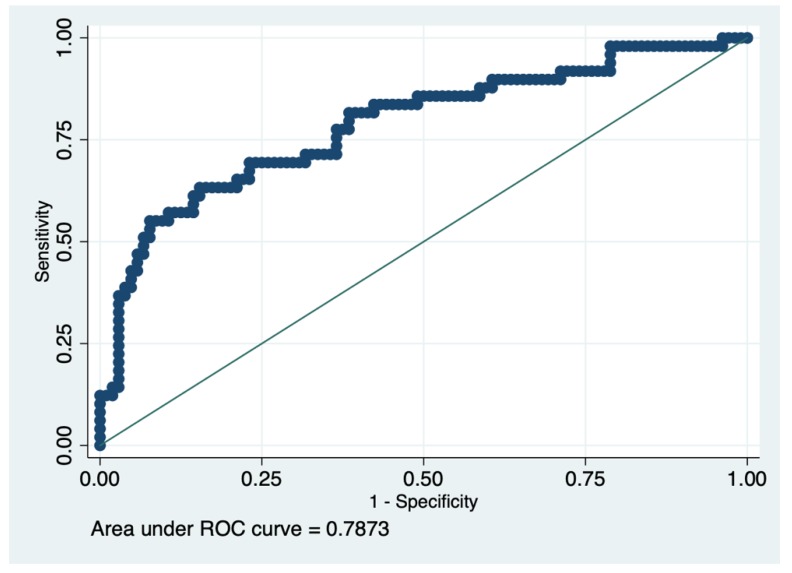
Area under the receiver operating characteristic (ROC) curve of the logistic model.

**Figure 2 jcm-08-00304-f002:**
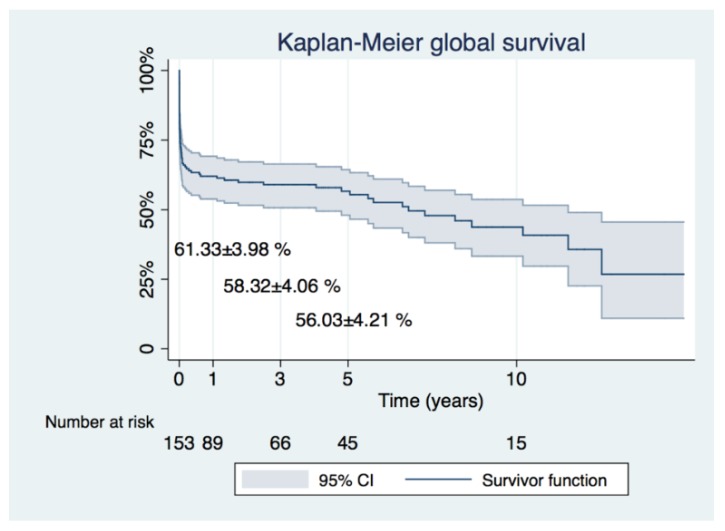
Kaplan–Meier curve depicting global survival of the entire cohort.

**Figure 3 jcm-08-00304-f003:**
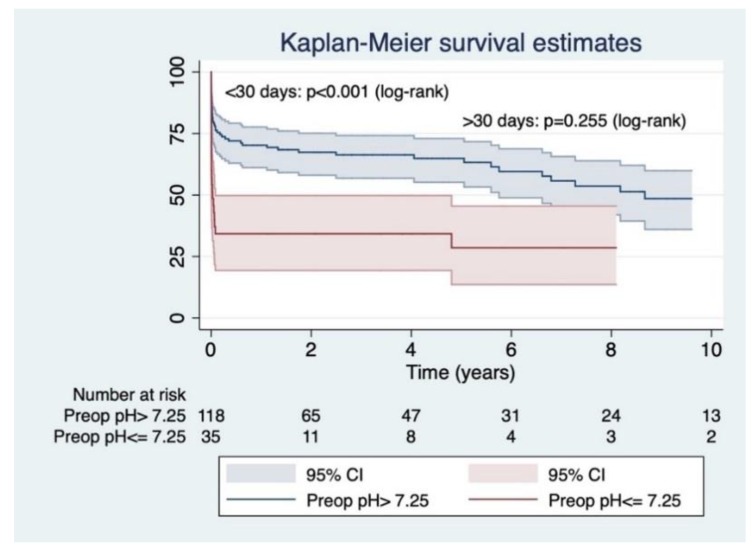
Kaplan–Meier curve for survival according to preoperative (prebypass) pH. A pH lower than 7.25 severely impacts early survival, but long-term (>30 days) survival is determined by other factors.

**Table 1 jcm-08-00304-t001:** Univariate preoperative risk assessment.

Variable	Survivors % (*n*)	Deceased % (*n*)	*p*
Age >65 years	18.2 (19)	36.7 (18)	0.01
Male sex	72.1 (75)	63.2 (31)	0.34
Risk factors			
Arterial hypertension	79.8 (83)	89.8 (44)	0.16
Diabetes	5.7 (6)	12.2 (6)	0.20
Body mass index	28 ± 4.9 (104)	29.92 ± 7 (49)	0.21
Marfan Syndrome	3.8 (4)	2.0 (1)	1.00
Bicuspid aortic valve	10.5 (11)	8.1 (4)	0.77
Redo procedure	6.7 (7)	2.0 (1)	0.43
Presentation			
Pericardial fluid >10 mm	27.8 (29)	53 (26)	<0.01
Severe aortic insufficiency	37.5 (39)	30.6 (15)	0.47
Aorta >5 cm	58.6 (61)	44.9 (22)	0.12
Ejection fraction	54 ± 6.4	51.6 ± 6.2	0.03
Entry site outside the ascending aorta	14.4 (15)	28.5 (14)	0.04
Malperfusion			
Penn non Aa	26.9 (28)	59.1 (29)	<0.01
Cardiogenic shock	10.5 (11)	36.7 (18)	<0.01
Limb ischemia	13.4 (14)	22.4 (11)	0.16
Renal ischemia	4.8 (5)	4 (2)	1.00
Cerebral ischemia	0.9 (1)	6.1 (3)	0.09
Mesenteric ischemia	0 (0)	6.1 (3)	0.03
Lab values			
Creatinine >1.7 mg/dL	22.1 (23)	40.8 (20)	0.02
Alanine aminotransferase (ALAT)	89.38 ± 200	205.73 ± 450.23	0.13
Aspartate aminotransferase (ASAT)	113.16 ± 276.54	216.59 ± 472.16	0.23
Acid base status			
Prebypass base deficit	3.63 ± 4.60	7.04 ± 6.70	<0.01
Prebypass pH	7.34 ± 0.08	7.26 ± 0.13	<0.01

**Table 2 jcm-08-00304-t002:** Multivariate risk analysis for early death.

Variable	Odds Ratio (OR)	VIF	*p*	95% Confidence Interval (CI)	
Creatinine> 1.7 mg/dL	1.28	1.31	0.59	0.51	3.22
Hypertension	1.70	1.14	0.40	0.47	6.08
Age> 65 years	2.4	1.06	0.06	0.96	5.96
Ejection fraction	0.96	1.09	0.33	0.90	1.03
Cardiogenic shock	3.30	1.48	0.03	1.10	9.82
Entry site	2.83	1.12	0.03	1.07	7.42
Pericardial effusion >10 mm	0.92	1.73	0.87	0.34	2.51
Prebypass base deficit	0.96	3.16	0.60	0.86	1.09
Prebypass pH×100	0.93	2.96	0.02	0.87	0.98
Malperfusion	1.77	1.17	0.25	0.66	4.74

**Table 3 jcm-08-00304-t003:** Cox regression analysis for early hazard phase (<30 days).

Variable	Hazard Ratio (HR)	*p*	95% CI	
Creatinine >1.7 mg/dL	1.09	0.80	0.53	2.24
Hypertension	1.40	0.54	0.46	4.25
Age >65 years	1.50	0.24	0.75	2.99
Ejection fraction	0.98	0.47	0.93	1.02
Cardiogenic shock	1.25	0.60	0.53	2.94
Entry site	2.10	0.04	1.00	4.42
Pericardial effusion >10 mm	1.37	0.44	0.60	3.09
Prebypass base deficit	0.99	0.83	0.91	1.07
Prebypass pH < 7.25	4.00	<0.01	1.64	9.75
Malperfusion	1.91	0.10	0.87	4.20

**Table 4 jcm-08-00304-t004:** Cox regression analysis for long-term (>30 days) hazard.

Variable	Hazard Ratio	*p*	95% CI	
Creatinine >1.7 mg/dL	2.42	0.13	0.76	7.72
Hypertension	0.96	0.94	0.29	3.12
Age >65 years	3.12	0.02	1.17	8.31
Ejection fraction	0.94	0.08	0.88	1.00
Cardiogenic shock	0.28	0.27	0.03	2.62
Entry site	0.84	0.82	0.18	3.82
Pericardial effusion >10 mm	0.93	0.91	0.27	3.21
Prebypass base deficit	0.88	0.06	0.78	1.00
Prebypass pH < 7.25	1.08	0.94	0.10	10.88
Malperfusion	1.40	0.51	0.50	3.92

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
