# Peer review of "How Much Malperfusion Is Too Much in Acute Type A Aortic Dissections?"

_jcm, 2019, doi:10.3390/jcm8030304_

Reviewer 1 Report

The authors looked at preop lactate acid levels > 6mmol/l as well as baseexcess -10mEq/l in a joined fashion integrating also preop serum pH. A total of 153 pts with acute TAAD were retrospectively analysed (collected over 15 years at one institution) which reflects about 10 pts per year.

Results: It would be important to tease out  a learning curve phenomenon (if there is) in view of the high overall early mortality of 32% and the recruiting time span of 15 years as things may have been altered and improved over such a long time span.

Table 2 and others:  Shouldn't Creatinin, Age etc be mentioned with a threshold?
Discussion: There is probably too much speculation on how to approach patients with acidosis in presence of clinical malperfusion syndromes; the recent Stanford data (ref. 21) prove that a quick response time and swift complete repair even out all malperfusion associated issues and result in excellent outcomes (and no sign. difference to patients with no malperfusion).

Data sets are relatively incomplete As only about half the cases had assessment of preop pH.

The conclusion and insinuation, that "compensation and correction" of preop parameters of ischemia/malperfusion/acidosis would ensure better survival is a hypothesis that needs to be tested prospectively eventually; but such statement is not yet justified on the basis of the presented data/results. Discussion needs major overhaul.

Author Response

Point 1: It would be important to tease out  a learning curve phenomenon (if there is) in view of the high overall early mortality of 32% and the recruiting time span of 15 years as things may have been altered and improved over such a long time span

Response 1: Thank you for your insight. There was clearly a learning curve involved as we operated all patients addressed to us at the beginning of our experience, between 2002-2008, regardless of their hemodynamic status and this has been now better explained in the revised manuscript (lines 246-248)

Point 2:Table 2 and others:  Shouldn't Creatinin, Age etc be mentioned with a threshold?

Response 2:I have now modified Table 1 in order to mention Creatinine, Age and Aorta size with a threshold. I’ve also revised the multivariate and survival analysis (Tables 2-4) to reflect that

Point 3:There is probably too much speculation on how to approach patients with acidosis in presence of clinical malperfusion syndromes; the recent Stanford data (ref. 21) prove that a quick response time and swift complete repair even out all malperfusion associated issues and result in excellent outcomes (and no sign. difference to patients with no malperfusion).

Response 3:It is unclear at the present moment how to address AAAD patients. While the Stanford group (formerly ref 21, now ref 13) have recently published excellent results from a surgery-first attitude, others (ref 10-12) have argued for the need to address the ischemia first, prior to aortic repair.  Our attitude has been similar to the Stanford group until recently, but their excellent results speak not only for their surgical skill, but also for the performance of their medical system. Some patients arrive to the hospital too sick to perform aortic surgery first, with an irreversible ischemia and trying to define those patients has been the focus of our study. It is now better explained in the Discussion section, which has been completely rewritten

Point 4:Data sets are relatively incomplete as only about half the cases had assessment of preop pH.

Response 4:Every patient had a measurement of preoperative pH and acid-base status performed just prior to surgery (n=153) and these values were used for the statistical analysis. However, in stable patients, the first set of acid-base analysis was performed on arrival (n=84). These data were not used for analysis, as they were incomplete. I’ve made this hopefully clear in the revised manuscript (lines 56-59) and by eliminating them from Table 1.

Point 5. The conclusion and insinuation, that "compensation and correction" of preop parameters of ischemia/malperfusion/acidosis would ensure better survival is a hypothesis that needs to be tested prospectively eventually; but such statement is not yet justified on the basis of the presented data/results. Discussion needs major overhaul.

Response 5:The Discussion section has been completely rewritten but I thoroughly agree with the Reviewer: a prospective study would be better suited to respond to the question: aorta-first or ischemia-first?

Kind regards for the time and effort put in revising this manuscript

Reviewer 2 Report

With their manuscript the authors address a clinically very relevant issue: does malperfusion have a relevant influence on the prognosis of a Type A Aortic Dissection?

Abstract:Comparing the conclusions of the abstract leads to questioning the relation to the manuscript title. In the conclusion the authors refer to the pH value and its relation to mortality and the authors prefer a two-time approach. Where remains here the relation to malperfusion and the title of the manuscript (line #2+3, line #25-27)?

Introduction: This section is not focused sufficiently: an orientation towards malperfusion in connection with the single region would be more useful. With this in mind 14 references are too much. It would improve the readability of the manuscript if the numbers “1” to “9” were presented spelled out (one, two,..) and from 10 onwards as numbers.“pH value” should be presented in a consistent fashion and not like in line #43 as “ph” and then again in line #135 “pH”.The end of an introduction should provide a hypothesis which is either proven or not.

Methods:If the abstract presents “(2) Methods” it should be shown for the sake of consistency as such in the manuscript and not “2. Experimental Section” (line #46). This whole section of the manuscript is extremely difficult to read and to interpret. Taken together it remains unclear how many subjects suffered from a malperfusion in which section. This should be presented in much more clear fashion. The question remains which group gets compared with which and how large is the final sample size? The reader needs to scrape this together from the text and the not very transparent tables.The definition of the single malperfusions is pretty unconventional (line #78-79). Why are thromboses visible on the CT scan not taken as criterion?

Statistical Analysis:A power calculation is even for a retrospective analysis possible and sensible to evaluate the relevance and the strength of the significant difference (line #90).

Results:In general, comments provided for the Methods section apply here as well. The definition of the sample size and the number of subjects is necessary. Furthermore, percentage figures should be presented with one digit (i. e. 17.4%). The 14-year-old subject with Marfan Syndrome should be excluded from the analysis since this would represent a very specific group (line #110/111).

In Section 3.1 the mixing of the sample becomes relevant: for the reader it is extremely difficult to recognize how many subjects belong to the group “malperfusion” (line #117-125).

Early Results: Here the problem is that the parameters were not defined earlier: for example – to assess the base deficit it would sensible to define the physiological value including range of variation (line #130). This applies in a similar fashion for the other assessed parameters. In general, the parameters should be closer associated with the context of malperfusion.

Table 1:This table is considered non-transparent: besides the percentage the absolute number of subjects should be presented in the groups. However, in the lower parts of the table it appears as if there are absolute values shown and not percentages.

Table 2:Why are there so many risk factors evaluated with this table, if the title of the manuscripts addresses malperfusion only. Furthermore, the relation of risk factors to malperfusion is not discernible.

Especially the localization of the malperfusion is of wider interest within the context of the focus of the manuscript than here distinguishable (line #156-162).

3.3 Intraoperative Data:Why are renal failures assessed? Under methods renal failure has been excluded and is not defined; therefore, it should not be evaluated in operative data.In long-term survival one cannot differentiate between life expectancy of reference group and subjects with malperfusion. Even set in context with Figure 3 this does not become any clearer.

Discussion:This section does not refer sufficiently to the title of the manuscript. Here it becomes obvious that there has been no hypothesis in the beginning which has to be proven or not. Referring to the title a clear statement on how much malperfusion is too much – from a philosophical point of view one would like ask: “too much for what?”

The authors managed to get together a relevant total sample based on the timeframe of the evaluation. The influence of medical advances over 15 years is discussed under limitations in a peripheral fashion. For example: did CT technique improve over the 15 years in a fashion that better data regarding dissection may be available?

At the end of the day the manuscript shows many little editing deficits like no unique formatting of the reference numbers in the text, additional commas and minor orthographic careless mistakes.
The subject of the manuscript title is without any doubt a clinically very significant issue. In the here presented form it does not meet the requirements of JCM and a publication cannot be supported.

Author Response

Response to Reviewer 2 Comments

Point 1: Comparing the conclusions of the abstract leads to questioning the relation to the manuscript title. In the conclusion the authors refer to the pH value and its relation to mortality and the authors prefer a two-time approach. Where remains here the relation to malperfusion and the title of the manuscript (line #2+3, line #25-27)? 

Response 1: Thank you for your comment. The conclusion of the abstract has been that a patient with a pH of 7.25 or less, due to global or local malperfusion, will have an unacceptably high mortality with an aorta-first technique.

Point 2: This section is not focused sufficiently: an orientation towards malperfusion in connection with the single region would be more useful. With this in mind 14 references are too much. It would improve the readability of the manuscript if the numbers “1” to “9” were presented spelled out (one, two,..) and from 10 onwards as numbers.

Response 2: Thank you for your suggestion. The Introduction section has been completely rewritten to better focus on malperfusion (lines 31-45). The references are formatted according to the Instructions for Authors on JCM’s webpage

Point 3: “pH value” should be presented in a consistent fashion and not like in line #43 as “ph” and then again in line #135 “pH”.

Response 3: This has now been corrected in the revised manuscript

Point 4: The end of an introduction should provide a hypothesis which is either proven or not.

Response 4: The Introduction has been rewritten and the scope of this study is now specifically spelled out (lines 31-45)

Point 5. If the abstract presents “(2) Methods” it should be shown for the sake of consistency as such in the manuscript and not “2. Experimental Section” (line #46). This whole section of the manuscript is extremely difficult to read and to interpret. Taken together it remains unclear how many subjects suffered from a malperfusion in which section. This should be presented in much more clear fashion. The question remains which group gets compared with which and how large is the final sample size? The reader needs to scrape this together from the text and the not very transparent tables.

Response 5: The Experimental Section of the manuscript has now been renamed Materials and Methods. The final sample size is specified in line 55. We’ve also rewritten the Malperfusion subsection (lines 116-122) as well as Table 1, to better present the preoperative local malperfusion syndromes.

Point 6: The definition of the single malperfusions is pretty unconventional (line #78-79). Why are thromboses visible on the CT scan not taken as criterion?

Response 6: The definitions of malperfusion are taken from the relevant literature. Malperfusion is a clinical and radiological entity. Thrombosis , for instance of the iliac vessels, does not automatically mean malperfusion: the patient may have a chronic occlusion of the iliac vessels, due to atherosclerosis and not acute  compression by the false lumen in AAAD. With regards to renal malperfusion, the medical literature is quite evenly split in this regard. We have included it in this revised manuscript

Point 7: A power calculation is even for a retrospective analysis possible and sensible to evaluate the relevance and the strength of the significant difference (line #90).

Response 7: In retrospective studies power analysis is only available as post-hoc analysis.

Point 8: In general, comments provided for the Methods section apply here as well. The definition of the sample size and the number of subjects is necessary. Furthermore, percentage figures should be presented with one digit (i. e. 17.4%). The 14-year-old subject with Marfan Syndrome should be excluded from the analysis since this would represent a very specific group (line #110/111).

Response 8: The number of subjects involved in the study was presented in line 55. Percentage figures are now presented with one digit after the comma throughout the manuscript. With regard to our youngest patient, aged 14, he had untreated aortic coarctation and not Marfan syndrome (line 112). Our sample represented consecutive patients operated between 2002-2017 so we included this patient and eliminating him would not modifiy the statistical results.

Point 9: In Section 3.1 the mixing of the sample becomes relevant: for the reader it is extremely difficult to recognize how many subjects belong to the group “malperfusion” (line #117-125).

Response 9: I’ve completely modified in text (lines 116-122), as well as in Table 1, the Malperfusion section. Hopefully, this is now better explained.

Point 10: Here the problem is that the parameters were not defined earlier: for example – to assess the base deficit it would sensible to define the physiological value including range of variation (line #130). This applies in a similar fashion for the other assessed parameters. In general, the parameters should be closer associated with the context of malperfusion.

Response 10: The acid-base variables (pH, base deficit) were analyzed as continuous data and these were the parameters used to measure malperfusion.

Point 11: Table 1. This table is considered non-transparent: besides the percentage the absolute number of subjects should be presented in the groups. However, in the lower parts of the table it appears as if there are absolute values shown and not percentages.

Response 11: I’ve completely overhauled Table 1 according to your suggestion: Groups are presented as percentages and absolute numbers, while continuous data, as those in the lower half of the table, are presented as mean±SD. It also includes a Malperfusion section, to better present the relevant ischemic territories.

Point 12: Table 2. Why are there so many risk factors evaluated with this table, if the title of the manuscripts addresses malperfusion only. Furthermore, the relation of risk factors to malperfusion is not discernible.

Especially the localization of the malperfusion is of wider interest within the context of the focus of the manuscript than here distinguishable (line #156-162).

Response 12: These were the risk factors that had a p<0.2 in the univariate analysis and were hence included in the logistic regression model. We weren’t able to analyze separately the location of the ischemia as independent variables as the numbers were too small: there were only 3 mesenteric ischemia patients (all perished), 4 cerebral ischemia patients, 7 renal ischemia and 25 patients with limb ischemia.

Point 13: Why are renal failures assessed? Under methods renal failure has been excluded and is not defined; therefore, it should not be evaluated in operative data.

Response 13: Renal malperfusion is now defined and included. I’ve hesitated a lot to do this, as the cause of creatinine increase may be due to preoperative renal impairment, however, coupled with a radiologic image of dissection or false lumen compression of the renal arteries, it is now considered  renal malperfusion. The literature itself is evenly split on this issue.

Point 14: In long-term survival one cannot differentiate between life expectancy of reference group and subjects with malperfusion. Even set in context with Figure 3 this does not become any clearer.

Response 14: Patients with malperfusion did not have an impaired early and late survival, compared to those without (see Table 2&3), which is why their survival was not graphically computed. On the other hand, patients with a preoperative pH<=7.25 had an impaired early survival but that effect wore out in the long term (please refer to Figure 3)

Point 15: This section does not refer sufficiently to the title of the manuscript. Here it becomes obvious that there has been no hypothesis in the beginning which has to be proven or not. Referring to the title a clear statement on how much malperfusion is too much – from a philosophical point of view one would like ask: “too much for what?”

Response 15: We attempted to find the most sensitive indices of malperfusion that would limit survival. So the question in the title: „How much malperfusion is too much”- refers to the question: „At what point the malperfusion becomes too severe and the ensuing mortality too high, and what markers do we have to assess that?”

Point 16 The authors managed to get together a relevant total sample based on the timeframe of the evaluation. The influence of medical advances over 15 years is discussed under limitations in a peripheral fashion. For example: did CT technique improve over the 15 years in a fashion that better data regarding dissection may be available?

Response 16: Of course that imaging modalities have evolved over 15 years, and it is one of the reason why our results have constantly improved. Better diagnosis and better emergency services leads to early repair with higher survival rates in this severe pathology.

Point 17: At the end of the day the manuscript shows many little editing deficits like no unique formatting of the reference numbers in the text, additional commas and minor orthographic careless mistakes.

Response 17: I’ve revised the manuscript for orthographic spelling mistakes and formatting.

Kind regards for the time and effort put in revising this manuscript.